# Enriched Red Wine: Phenolic Profile, Sensory Evaluation and In Vitro Bioaccessibility of Phenolic Compounds

**DOI:** 10.3390/foods12061194

**Published:** 2023-03-11

**Authors:** Óscar A. Muñoz-Bernal, Alma A. Vazquez-Flores, Laura A. de la Rosa, Joaquín Rodrigo-García, Nina R. Martínez-Ruiz, Emilio Alvarez-Parrilla

**Affiliations:** 1Departamento de Ciencias Químico-Biológicas, Instituto de Ciencias Biomédicas, Universidad Autónoma de Ciudad Juárez, Anillo Envolvente del Pronaf s/n, Fovisste Chamizal, Ciudad Juárez C.P. 32300, Mexico; 2Departamento de Ciencias de la Salud, Instituto de Ciencias Biomédicas, Universidad Autónoma de Ciudad Juárez, Anillo Envolvente del Pronaf s/n, Fovisste Chamizal, Ciudad Juárez C.P. 32300, Mexico

**Keywords:** red wine, phenolic compounds, bioaccessibility, sensory profile, phenolic profile

## Abstract

The beneficial health effect of red wine depends on its phenolic content and the phenolic content in red wines is affected by ecological, agricultural, and enological practices. Enriched wines have been proposed as an alternative to increase the phenolic content in wines. Nevertheless, phenolic compounds are related to the sensory characteristics of red wines, so enrichment of red wines requires a balance between phenolic content and sensory characteristics. In the present study, a Merlot red wine was enriched with a phenolic extract obtained from Cabernet Sauvignon grape pomace. Two levels of enrichment were evaluated: 4 and 8 g/L of total phenolic content (gallic acid equivalents, GAE). Wines were evaluated by a trained panel to determine their sensory profile (olfactive, visual, taste, and mouthfeel phases). The bioaccessibility of phenolic compounds from enriched red wines was evaluated using an in vitro digestive model and phenolic compounds were quantified by High Performance Liquid Chromatography coupled to tandem mass spectrometry (HPLC-MS/MS). Enrichment increased mainly flavonols and procyanidins. Such an increase impacted astringency and sweetness perceived by judges. This study proposes an alternative to increase the phenolic content in wines without modifying other main sensory characteristics and offers a potential beneficial effect on the health of consumers.

## 1. Introduction

Winemaking and wine consumption are activities that have accompanied humanity for centuries [1]. Numerous studies demonstrate that moderate consumption of wine and its polyphenolic compounds can benefit the health of consumers [2], reducing the risk of suffering from chronic diseases such as diabetes mellitus, metabolic syndrome [3], and cardiovascular diseases [1,4]. Polyphenolic compounds are the most abundant class of compounds that constitute the non-alcoholic portion of red wine. They are classified according to their structure into two large classes: flavonoids and non-flavonoids. In wine, the non-flavonoid fraction is represented by hydroxybenzoic acids (gallic, syringic, and protocatechuic acids), hydroxycinnamic acids (caffeic and coumaric acids), and stilbenes (resveratrol and piceatannol) [2]. Among flavonoids, wine contains anthocyanins (malvidin), flavonols (quercetin), flavanones, flavones, flavan-3-ols, and proanthocyanidins in their aglycone or glycosylated form [3,4].

Currently, one of the best-marketed wines is Merlot [5], which contains moderate levels of phenolic compounds. Nevertheless, the quantity and variety of phenolic content in red wines depend on the climatic, geographical, and agricultural conditions of grape production and origin, and even on the different phases of winemaking [2,6], so it is not constant for any given wine. For this reason, improving the functional quality of wine through the addition of polyphenolic compounds is one of the priorities of the wine industry [7]. In this sense, some authors have applied different tools, such as exposing crops to UV light radiation to increase the amount of resveratrol and piceatannol [8]. Others have directly added commercial extracts of grapevine shots [9] or extracts of catechins from green tea and grape seeds to the final product [10], as well as pure extracts of trans-resveratrol [11] and zein nanoparticles [3]. Although these tools can enrich wines, they also have important limitations such as the lack of control in the production of polyphenolic compounds, the enrichment of only specific compounds, and the development of unwanted organoleptic characteristics in the final product [3,11]. The use of grape pomace (small stalks, skins, and grape seeds), a by-product of winemaking, has been also reported for obtaining an enriched wine with better antioxidant characteristics. This alternative has the advantage of allowing the reuse of the industry by-products in addition to increasing the possible positive effects on the health of the wine consumer [2,7]. Grape pomace is an important source of polyphenolic compounds, and its valorization has been studied to reduce waste management costs and obtain valuable products for cosmetic, pharmaceutical, and food industries [12,13] It has been reported that extracts obtained from grape skins and seeds present high antioxidant activity [14]. On the other hand, encapsulated polyphenolic compounds extract obtained from grape pomace has shown increase antioxidant activity and reduce polyphenolic compounds degradation [12].

In wines, sensory evaluation plays a very important role since it analyzes important attributes in wines such as color, clarity, odor, taste, and astringency, among others. However, these sensory characteristics can be modified by the presence of polyphenolic compounds and their low solubility [11]. Achieving a red wine with a good organoleptic balance is of great interest. The digestive system presents a wide variety of conditions (pH, enzymes) and molecules such as carbohydrates and proteins, which can form covalent and non-covalent interactions with the phenolic compounds, reducing their digestibility, bioaccessibility, and bioavailability [15]. Thus, the main aim of this study was to enrich a red wine with phenolic compounds from grape pomace in controlled conditions to increase its functional quality and evaluate the sensory attributes of the wines, as well as to quantify the bioaccessibility of the added compounds throughout an in vitro digestive system. 

## 2. Materials and Methods

### 2.1. Chemicals

Phenolic compound standards: gallic acid, syringic acid, protocatechuic acid, caffeic acid, chlorogenic acid, naringenin, epicatechin, catechin, hesperetin, resveratrol, quercetin, rutin, ellagic acid, myricetin, kaempferol, luteolin, isorhamnetin, epigallocatechin, gallocatechin, gallocatechin gallate, epicatechin gallate, procyanidin A1, procyanidin A2, procyanidin B1, procyanidin B2, procyanidin C1, and corilagin were HPLC grade and purchased from Merck (St. Louis, MO, USA). Acetonitrile, formic acid, methanol, hexane, and water were HPLC/spectrum grade and purchased from Tedia (Fairfield, OH, USA). All other reagents and solvents used were analytical grade from Merck (St. Louis, MO, USA).

### 2.2. Red Wines and Grape Pomace Samples

Red wine samples from *Vitis vinifera* L cv. Merlot and grape pomace from Cabernet Sauvignon were used in the present study. All samples were kindly donated by the Grupo Alximia S.A. de C. V. winery, located in Guadalupe Valley, Baja California, Mexico. This region, located in the northwestern of Baja California, Mexico, is an intermountain valley with a semi-arid Mediterranean climate. The average annual rainfall is 298 mm/year with an average annual temperature of 17.9 °C [16]. Red wine samples were obtained from the 2017 vintage after 1 year of barrel storage and bottling in 750 mL dark glass bottles with plastic cork plugs. Grape pomace samples were obtained from the 2018 vintage. Winemaking was performed with daily pumping over and cold maceration for 14 days. After this process, wine was pressed, and grape pomace was obtained. Cabernet Sauvignon grape pomace samples were collected just after the pressing process and vacuum stored at −20 °C until transportation to the laboratory. Grape pomace sample was taken from the total production of the winery, a total of 100 kg were taken from different parts of the pressed grape pomace to asses a representative sample. Samples were transported under controlled temperature to the laboratory at Universidad Autónoma de Ciudad Juárez, located in Ciudad Juárez, Chihuahua, México. 

### 2.3. Phenolic Compound Extraction

Fresh grape pomace presented a total moisture of 61.51%. Grape pomace samples were oven-dried at 55 °C (Isotemp oven, Fischer Scientific^®^, Waltham, MA, USA) for 72 h until constant weight. Once dried, samples were ground and sieved (120 µm mesh). Dried grape pomace was placed into an industrial mixer to homogenize the whole sample batch. Then samples were vacuum stored into individual bags of 1 kg each until further analysis. Phenolic compounds from grape pomace were extracted according to Muñoz-Bernal et al. [4] with slight modifications. Briefly, dried grape pomace was defatted with hexane in a 1:5 proportion (solid: solvent) by ultrasound (B5000, Branson^®^, Brookfiled, CT, USA) for 10 min. The solid phase was separated by centrifugation (Sorvall 16R, Thermo Scientific^®^, Waltham, MA, USA) at 3500 rpm at 4 °C for 10 min. The procedure was repeated using the same solvent proportion. Defatted grape pomace was oven-dried at 40 °C for 24 h. Phenolic compound extraction from defatted grape pomace was carried out with 40% ethanol in water (*v*:*v*). Defatted grape pomace was mixed with extraction solvent in a ratio of 1:5 (solid: solvent), then the mixture was extracted by ultrasound for 30 min at room temperature and separated by centrifugation at 2000 rpm for 15 min at 4 °C. The liquid phase was collected and stored at 4 °C, the solid phase was reextracted under the same conditions. Both liquid phases were mixed, and ethanol was distilled in a rotary evaporator (R-4, Büchi^®^, Flawil, Switzerland) at 45 °C. The aqueous part was freeze-dried (Freezone 6, Labconco^®^, Kansas City, MO, USA). The final powder of grape pomace extract (GPE) was vacuum stored at −80 °C until further analysis. 

### 2.4. Phenolic Compound Characterization by Spectrophotometric Methods

For spectrophotometric analysis, grape pomace extract (GPE) was diluted at 2 mg/mL in 14% ethanol (*v*:*v*). Wine samples were used directly or diluted with 14% ethanol. Total phenolic content (TPC) from GPE and wine samples was determined by the Folin–Ciocalteu method, as described by Muñoz-Bernal et al. [6]. Results were expressed in milligrams of gallic acid equivalents per gram of GPE (mg GAE/g GPE) or liter of wine (mg GAE/L). Total flavonoids (TF) were quantified by the aluminum complexation method, as described by Muñoz-Bernal et al. [6]. Results were expressed in milligrams of catechin equivalents (CE) per gram of GPE (mg CE/g GPE) or liter of wine (mg CE/L). Total anthocyanins (TA) were determined by the pH shift method, according to Lee et al. [17]. Results were expressed as milligrams of malvidin 3-glucoside equivalents (M3E) per gram of GPE (mg M3E/g GPE) or per liter of wine (mg M3E/L). Condensed tannins or procyanidins were quantified by the *p*-dimethylaminocinnamaldehyde (DMAC) method according to Muñoz-Bernal et al. [6]. Results were expressed as milligrams of catechin equivalents per gram of GPE (mg CE/g GPE) or per liter of wine (mg CE/L). All analyses were performed in triplicate. 

### 2.5. Phenolic Compounds Characterization by HPLC-ESI-QTOF-MS/MS

The phenolic profile from GPE and wine (control and enriched) was determined by HPLC-MS/MS according to Muñoz-Bernal et al. [4]. Samples (500 µL) were filtered using a Nylon syringe filter 13 mm, 0.45 µm (Titan 3, Thermo Scientific^®^, Waltham, MA, USA). Samples were analyzed in a 1290 Infinity series (Agilent Technologies, Inc., Santa Clara, CA, USA) high-performance liquid chromatography (HPLC) system model using a 1290 Infinity quaternary pump with a built-in degasser, 1290 Infinity autosampler with temperature control, 1290 Infinity thermostated column compartment, and 1290 Infinity diode array detector. A ZORBAX^®^ (Agilent Technologies, Inc., Santa Clara, CA, USA) C_18_ column (50 mm × 2.1 mm, 1.8 µm) was used at 25 °C for separations. The mobile phase was: 0.1% formic acid in water (A) and acetonitrile 100% (B). The gradient elution was: 0–1 min 10% B, 1–4 min 30% B, 4–6 min 38% B, 6–8.5 min 60% B, 8.5–10 min 10% B. The flow rate was 0.4 mL/min with an injection volume of 3 µL. The mass spectrometer system was an Agilent 6530 Accurate-Mass QTOF MS/MS equipped with an electrospray ionization (ESI) operated in negative mode. Conditions were as follows: nitrogen was used as a drying gas at 340 °C, with a flow rate of 13 L/min, nebulizer gas pressure 30 psi, capillary voltage 4000 V, fragmentor voltage 175 V, skimmer voltage 65 V, and m/z scan range 100–1100 for MS and 100–1000 MS/MS. The phenolic compounds identification was carried out using the software Mass Hunter Qualitative version B.07.00., according to the methodology described by Muñoz-Bernal et al. [18]. 

The quantification of phenolic compounds was performed with external calibration curves, from available phenolic standards, using the mass peak areas extracted from the ion chromatogram [18]. Calibration curves were prepared by dilution from stock solutions in methanol (1 mg/mL each compound). Range concentrations were from 0.0007 mg/mL to 0.1 mg/mL. For those compounds whose reference was not available, a calibration curve from structurally related compounds was used [19]. The limit of detection (L.O.D. = 3 times signal to noise ratio (S/N); S/N = 12,525.6) and the limit of quantitation (L.O.Q. = 10 times S/N) were calculated for each standard compound and are presented in Appendix A. All samples were analyzed in triplicate, and results were expressed in mg/g GPE for extracts or mg/L for wines.

### 2.6. Wine Enrichment

The enrichment process was carried out at 25 °C and light absence in the laboratory. To enrich red wines, sealed bottles were opened and approximately 100 mL were poured into a beaker. Different amounts of GPE (523.75 mg/g extract) were added to assess a final concentration of 4.23 and 7.89 g GAE/L (W4 and W8, respectively) enrichment, then the mixture was returned to the bottle. Bottles were sealed with new plastic cork plugs and mixed to assure GPE solubility. Enrichment was performed one week before analysis. Before analyses, bottles were filtered using coffee filters (Conaissur^®^ 8–12 cup basket, Item #7712, Rockline industries, Sheboygan, WI, USA) to eliminate any precipitate formed. Total phenolic content by spectrophotometric method was performed to ensure the final concentration in enriched wines.

### 2.7. Sensory Profile 

The control red wine (without enrichment WC), W4, and W8 were sensory characterized with a descriptive analysis by a trained panel of 10 judges according to the method described by Lawless and Heymann [20]. Attributes in the visual phase were color and brightness. Attributes in the olfactive phase were odor intensity and descriptors determined by a focus group technique. Descriptors determined by the panel were alcohol, prune, wood, caramel, fermented, and fruit. In the oral phase, mouthfeel attributes were body wine and astringency; and taste attributes were salty, sourness, bitterness, and sweetness. All tests were conducted in individual booths and the judges used a 150 mm linear scale, labeled at the ends as “Not at all…” and “Extremely…” for each attribute or descriptor. Each judge was provided with 10 mL of wine placed in 4 oz black plastic glasses, except for the visual phase, where 4 oz transparent plastic glasses were used. Samples were identified with three-digit random numbers and presented to the judges in a balanced, counterbalanced, and randomized order along with evaluation sheets. Judges rinsed their mouths with purified water at the beginning and between samples for the oral phase, and the samples were spat out after tasting them. The judges were asked not to eat, drink, smoke, or take any oral product one hour before tasting. Two attributes or descriptors were evaluated per session of 60 min. Standards for each attribute or descriptor were used at the beginning of the test and each test was performed by duplicate. The standards used can be consulted in Appendix A.

### 2.8. Simulated In Vitro Gastro-Intestinal Digestion (In Vitro Bioaccessibility of Phenolic Compounds)

A simulated human digestive system was prepared according to Kopf-Bolanz et al. [21] and Vazquez-Flores et al. [22] with some modifications. This system consisted of continuous batch series mimicking oral, gastric, and intestinal juices (salts, enzymes, and pH) similar to those found in the human gut. Mouth stage: 2.25 mL of wine sample was mixed with 3 mL of saliva juice (KCl 0.93 mg/mL, KSCN 0.08 mg/mL, KH_2_PO_4_ 2.72 mg/mL, NaHCO_3_ 0.67 mg/mL, NaCl 0.24 mg/mL, MgCl_2_ (H_2_O)_6_ 0.06 mg/mL, CO(NH_2_)_2_ 0.04 mg/mL, CaCl_2_ (H_2_O)_2_ 0.29 mg/mL, mucin II 1 mg/mL, lysozyme 0.02 mg/mL and saliva α-amylase 4 µg/mL and pH 6.8) and placed in a water bath shaker at 37 °C and 100 rpm for 5 min. After this time, 500 µL of the sample was taken and stored at −80 °C until further analysis. Gastric stage: 6 mL of gastric juice (KCl 2.62 mg/mL, KH_2_PO_4_ 0.12 mg/mL, NaCl 2.4 mg/mL, CO(NH_2_)_2_ 0.01 mg/mL, MgCl_2_ (H_2_O)_6_ 0.12 mg/mL, NaHCO_3_ 2.18 mg/mL, NH_4_Cl 0.05 mg/mL, glucuronic acid 0.02 mg/mL, glucosamine 0.33 mg/mL, galactose 0.32 mg/mL, CaCl_2_ (H_2_O)_2_ 0.09 mg/mL, mucin II 1.4 mg/mL, bovine serum albumin (BSA) 1 mg/mL and pepsin 2.5 mg/mL and pH 1.3) were added to the sample and mixed, then the sample was placed in a water bath shaker at 37 °C, 100 rpm for 2 h. After this period, 500 µL of the sample was taken and stored at −80 °C until further analysis. For the intestinal stage, 6 mL of pancreatic juice (KCl 0.5 mg/mL, KH_2_PO_4_ 0.11 mg/mL, NaHCO_3_ 3.61 mg/mL, NaCl 1.92 mg/mL, CO(NH_2_)_2_ 0.11 mg/mL, MgCl_2_ (H_2_O)_6_ 0.07 mg/mL, CaCl_2_ (H_2_O)_2_ 0.09 mg/mL, mucin III 1.4 mg/mL, BSA 1 mg/mL, pancreatin 18 mg/mL) and 3 mL of bile juice (KCl 0.5 mg/mL, KH_2_PO_4_ 2.42 mg/mL, NaHCO_3_ 1.6 mg/mL, NaCl 1.92 mg/mL, CO(NH_2_)_2_ 0.23 mg/mL, MgCl_2_ (H_2_O)_6_ 0.07 mg/mL, NaH_2_PO_4_ 3.75 mg/mL, CaCl_2_ (H_2_O)_2_ 0.54 mg/mL, BSA 1.8 mg/mL, bile 60 mg/mL) were added to the sample and it was incubated at 37 °C, 100 rpm for 2 h. After this period, 500 µL of the sample was taken and stored at −80 °C until further analysis. 

Samples coming from all simulated digestive stages (oral, gastric, and intestinal) were further dialyzed. The samples were transferred to 50 mL plastic tubes. A dialysis bag containing a blank solution consisting of a 3 mL mixture of saliva, gastric, pancreatic, and bile juices (without enzymes and phenolic compounds) was submerged into the tube and was allowed to dialyze for 4 h at 37 °C with agitation at 100 rpm. The solution inside the bag represented the dialyzable fraction and the potential bioavailable compounds, meanwhile the solution outside represented the non-dialyzable fraction and the potential colon available compounds. Three independent experiments were performed for each sample, and three blanks (water instead of wine) were performed to avoid interferences of digestion reagents. Samples from each digestive stage were centrifugated at 14,000 rpm for 10 min. Then, samples were filtered using 0.45 µm Nylon syringe filters and analyzed by HPLC- MS/MS according to the previously described method. 

### 2.9. Statistical Analysis 

Results from spectrophotometric and chromatographic techniques are expressed as mean ± standard deviation (SD) of three replicates. One-way analysis of variance (ANOVA) was performed, and Fisher LSD was used for the comparison of mean values. Both were performed at a significance *p* < 0.05. For sensory analysis, repeated measures ANOVA was performed, and Fisher LSD was used for the comparison of mean values at a 0.05 significance level. All the statistical analyses were performed using XLSTAT version 2022.4.1 (Addinsoft^®^, Paris, France).

## 3. Results and Discussion

### 3.1. Wine and Grape Pomace Phenolic Characterization

The understanding of initial conditions from wine and grape pomace on phenolic content and phenolic fractions, such as flavonoids, condensed tannins, and anthocyanins, is important to determine the main fractions that can be modified by enrichment. For this reason, before preparing the enriched red wines, the phenolic fractions of Cabernet Sauvignon grape pomace extract (GPE), and Merlot red wine without enrichment (WC) were quantified by spectrophotometric methods. Results are shown in Table 1.

As observed in Table 1, WC presented a total phenolic content of 2310 mg GAE/L. On the other hand, GPE presented 523 mg GAE/g extract. In both samples, flavonoid content was higher than condensed tannin and monomeric anthocyanins content.

Merlot wines showed a higher content of flavonols than flavan-3-ols, in agreement with previously reported results [23]; this is in accordance with results obtained in the present study for Merlot wine (Table 1). According to Casassa et al. [24], in Merlot wines, extraction of condensed tannins from grape seeds starts after 20 days of maceration. In the present study, the maceration process was performed for 14 days, which may explain the lower condensed tannins content. 

Grape pomace contains phenolic acids, flavonols, flavan-3-ols, and anthocyanins [25]. The extraction process is critical to obtain phenolic compounds from grape pomace. The drying process modifies the phenolic compounds that can be extracted from grape pomace [20]. In a study conducted by Goula et al. [26], the authors observed that temperatures over 60 °C reduced the phenolic content of grape pomace extracts. For this reason, in this study, grape pomace was dried at 55 °C to prevent phenolic compound degradation. Moreover, oven drying was chosen over freeze drying because it may be a suitable condition within the wine industry and has a lower cost. The solvent proportion also plays an important role in phenolic extraction. It has been reported that the use of ethanol in a ratio of 20:1 (solvent: solid) using ultrasound extraction assisted exhibits better extraction yields than conventional extraction strategies [27]. In the present study, the use of 40% ethanol in a 1:5 (solid: solvent) proportion exhibited the best phenolic compound extraction. In the GPE, monomeric anthocyanin content was 1.67 mg M3E/g extract (Table 1). According to Drosou et al. [28], anthocyanins are thermally sensitive and temperatures of 55 °C can reduce anthocyanin degradation. This could explain the low anthocyanin content in the GPE.

### 3.2. Wine Enrichment

According to results observed in Table 1, the addition of GPE can increase the total phenolic content and the flavonoid and condensed tannin content of wine samples. Two levels of wine enrichment were established: 4 and 8 g GAE/L. For enrichment, 3.67 g of GPE were added to sample W4 (real concentration 4. 23 g GAE/L) and 10.55 g of GPE were added to sample W8 (real concentration 7.83 g GAE/L).

To observe the effect of enrichment on the phenolic profile of wines, individual phenolic compounds were identified and quantified using HPLC-MS/MS. The phenolic profile from GPE was also evaluated. A total of 14 hydroxybenzoic acids, 5 hydroxycinnamic acids, 8 stilbenes, 2 phenylethanoids, 2 coumarins, 4 flavones, 7 flavanones, 16 flavonols, and 13 flavan-3-ols were identified in samples. The phenolic profile of WC, W4, W8, and GPE is presented in Table 2. Spectral information can be consulted in Appendix A.

Moderate wine consumption is related to the prevention of cardiovascular diseases [29]. However, the beneficial effect of red wine on health depends on its phenolic content and phenolic profile [30]. The enrichment of wines has been performed mainly by the addition of specific phenolic compounds like resveratrol [8,29]. In contrast, the enrichment strategy used in the present study promotes the use of grape pomace, the principal by-product of the wine industry. Moreover, grape pomace is an important source of phenolic compounds that can be exploited to increase phenolic content in red wines. The GPE obtained contains mainly flavanols and flavan-3-ols, not only a specific group of phenolic compounds. The total phenolic compound content quantified by HPLC-MS/MS can be observed in Table 2. It can be observed that the value obtained from HPLC-MS/MS was lower than the value obtained by spectrophotometric methods. This difference can be explained since spectrophotometric results are unspecific. On the other hand, a low concentration of phenolic compounds determined by HPLC-MS/MS can be attributed to those phenolic compounds that cannot be identified or quantified. 

GPE presented mainly flavonols such as quercetin, syringetin, and their respective derivative compounds (Table 2). This is consistent with the phenolic profile previously reported for Cabernet Sauvignon GPE [31]. Hydroxybenzoic acids were more abundant than hydroxycinnamic acids. Gallic acid was present in higher content compared with syringic, ellagic acids, and pyrogallol. In contrast, caffeic acid was present in sample but was found under the limit of quantification. The same trend was observed for methyl-ferulate. Phenolic acids identified in GPE (Table 2) are in agreement with other authors [32,33]. On the other hand, stilbenes were not identified in the GPE sample. Stilbenes can be found mainly in the grapevine canes [34] and not in the grape skins; this may explain the absence of stilbenes in the GPE. The grape pomace used in this study is constituted of skins, seeds, and short stems. Seeds and stems are known to contain flavan-3-ols such as catechins and procyanidins [25]. The presence of catechin, epicatechin, and procyanidins were confirmed in the obtained GPE (Table 2). Moreover, GPE was diverse in procyanidins, GPE exhibited the presence of monomers B3 and B3 and trimers C1 and C2. The content of flavan-3-ols in GPE is similar to those previously found [31]. 

The phenolic profile from Merlot red wine (WC) (Table 2) indicates that gallic and syringic acids were higher than protocatechuic and ellagic acids. Gallic acid is reported as the main hydroxybenzoic acid in Merlot wines [35,36]. Caffeic acid and methyl-ferulate were the main hydroxycinnamic acid present in WC. The main hydroxycinnamic acids reported in wines are hydroxycinnamoyl tartaric acids such as caftaric, coutaric, and fertaric acid [37]. Such compounds were not identified in WC. The low content of *p*-coumaric acids has been reported previously for red wines, since *p*-coumaric acid is associated with anthocyanin derivatives for color stabilization [37]. Stilbenes content in WC was low; most of the compounds from this family were found under the limit of quantification. Resveratrol content in red wines depends on several factors; concentrations from 0.22 to 1.75 mg/L have been reported for Merlot wines [38]. The most common flavonols reported for wines are quercetin, myricetin, laricitrin, syringetin, isorhamnetin, and kaempferol [35,36,37]. In WC, flavonols were present mainly in their glycosylated form rather than as free aglycones (Table 2). Interestingly, free aglycones of flavonols were found in GPE. This can suggest that during the fermentation-maceration process, the hydrolysis of flavonols glycosides may occur [37]. 

It has been proposed that berry size and seed content is determinant of flavan-3-ols content in red wines. In this sense, Merlot grapes are characterized by a small berry size and high seed content that may lead to the high concentration of flavan-3-ols into the wine [39]. Table 2 describes the content of flavan-3-ols in WC. Catechin, epicatechin, and procyanidin B1 and B2 presented a higher content compared to gallocatechin and epicatechin-3-glucuronide. These results are in agreement with previous studies that reported catechin, epicatechin, and procyanidin dimers as the main flava-3-ols present in Merlot wines. The individual content of flavan-3-ols reported previously [36,37] is higher than those observed in the present study. 

The phenolic profiles of W4 and W8 were different compared to WC (Table 2). For hydroxybenzoic acids, protocatechuic, syringic, and ellagic acid increased their content after the enrichment. Pyrogallol and 4-methyl gallic acid were only found in W4 and W8 samples; both compounds were incorporated by the GPE. In the case of hydroxycinnamic acids, caffeic acid and methyl ferulate increased their content. In previous study, the enrichment of red wine with a GPE also exhibited an increase in gallic and syringic acids [3]. In contrast, gallic content decreased in W4 (10.51 mg/L) and W8 (4.90 mg/L) samples compared to WC (12.05 mg/L). This behavior can be attributed to a precipitation effect since gallic acid was the only compound that presented a decrease instead of an increase in content. In general terms, the total amount of phenolic compounds increased as the GPE increased (Table 2).

The stilbene profile of W8 was different from WC and W4 (Table 2). Resveratrol was quantified only in the W8 sample (11.32 mg/L). Resveratrol is described as being mainly responsible for the beneficial effect of moderate red wine consumption [40]. For this reason, several studies are focused on increasing resveratrol and other stilbenes such as piceid, astringin, and piceatannol in red wines. Diverse techniques have been applied to increase stilbenes in red wines. The exposure of grapes to ultraviolet C light (UVC) have shown increased resveratrol in wines from 0.91 to 3.07 mg/L [8]. The use of a commercial grapevine shoot extract with stilbenes has been demonstrated to increase piceatannol considerably from 5 mg/L to 80 mg/L. Moreover, resveratrol oligomers (dimer and tetramers) increase their content in enriched wines [9]. As can be observed, the increase of stilbenes depends on the method for enrichment. The enrichment of stilbenes reached in W4 and W8 is lower compared to when grapevine shoot extract was used, but higher compared with UVC light treatment. Nevertheless, the enrichment proposed in the present study is not targeted only for stilbenes.

According to spectrophotometric results, flavonoids were the main fraction present in GPE and transferred to the enriched wines. This was confirmed in the phenolic profile of W4 and W8 (Table 2). In general terms, almost all phenolic compounds increased with the addition of GPE in a dose-dependent manner. This was more evident for flavonoids. Flavonols such as kaempferol, kaempferol-3-glucoside, and myricetin, present in GPE, were only detected in W8. Flavonols are more abundant in grapes with thick-skinned fruits such as Cabernet Sauvignon [41]. This may explain the high content of flavonoids obtained in enriched wines.

Among flavonoids, flavan-3-ols were one of the main flavonoid fractions present in GPE, and consequently, they increased in dose-dependent manner in enriched wines. W8 presented the highest concentration of monomeric flavan-3-ols (catechin and epicatechin) and procyanidins (B1 and B2) compared to WC and W4. In a previous study, the enrichment of red wine with a GPE also presented an increase in catechin, epicatechin, as well as procyanidin dimers and timers [3].

### 3.3. Sensory Profile

Phenolic compounds content in red wines is strongly correlated with their sensory characteristics such as color, bitterness, and astringency [42]. Considering that the enrichment of red wine modified the different phenolic fractions and phenolic profile, sensory evaluation was performed to determine the influence of these changes on their sensory profile. For a better understanding and data analysis, the intensity linear scale (150 mm) was divided into 5 sections: low (L, 0–37 mm), medium-low (ML, 38–74 mm), medium (M, 75 mm), medium-high (MH, 76–112 mm), and high (H, 113–150 mm). Results from the descriptive sensory evaluation are shown in Figure 1.

General odor intensity results (Figure 1a) indicate that WC and W4 (68.9 and 69.9 mm, respectively) were ubicated in the ML scale; meanwhile W8 was rated at the MH scale (79.3 mm). However, the general odor from enriched red wines showed no significant difference from WC. According to these results, the enrichment of red wine with GPE does not modify the general odor of enriched red wines. Wine descriptors determined by panelists in the olfactive phase were alcohol, prune, wood, caramel, fermented, and fruit as the more representative in samples. For the alcohol descriptor, all samples were rated as ML and were not modified by GPE enrichment. As for prune, WC and W4 were rated as ML and W8 in MH. As observed in Figure 1a, prune descriptor rates increased as GPE enrichment increased. Nevertheless, no significant difference was observed indicating that enrichment in W8 was not sufficient to modify the prune intensity perceived by panelists. The wood odor in all wine samples was rated as ML; scores ranged from 49.9 to 59.4 mm. In caramel odor, samples were rated in MH and presented a similar behavior to prune odor intensity. This descriptor was perceived as follows WC < W4 < W8, though no significant difference was observed in enriched red wines compared to the control. For fermented, WC was rated in ML (71. 8 mm) and W4 and W8 samples were rated at MH (82.9 and 86 mm, respectively), although these differences were not significant. Finally, fruit intensity was rated at ML in all wine samples. In general, no differences were observed in odor intensity and odor descriptors of wines enriched with GPE. Visual attributes evaluated in wine samples were color and brightness. According to Figure 1a, color was rated H in all samples. On the other hand, brightness was rated in ML for all samples.

The taste attributes analyzed in wine samples were sourness, bitterness, salty, and sweetness (Figure 1b). Bitterness, sourness, and salty were scored at ML for all wine samples. Data from Figure 1b shows that for salty, bitterness and sourness increased as GPE enrichment was increased. However, such an increase in W4 and W8 was not statistically significant. In contrast, the sweetness of the W8 (18.5 mm) was perceived as less compared to WC (30.3 mm). For this work, astringency and body wine were the mouthfeel attributes studied (Figure 1b). Enriched red wines W4 and W8 were more astringent with significant differences to WC. However, statistically, no significant difference was observed between W4 and W8.

General odor and odor descriptors were not modified in enriched wines (W4 and W8) compared with control wine (Figure 1). Oxygen diffusion into wine favors enzymatic oxidations that help aromas release. This process can occur during maceration but also during wine bottling when oxidation, polymerization, and complexation reactions take place in wine [24,43]. The addition of 200 mg/L of resveratrol was reported to modify odor intensity compared to the control wine [11]. The results found in the present study are contrary to previous studies, where the process of enrichment modified their odor intensity. These differences can be attributed to the method of enrichment and time of GPE addition. The timing of bottling has been reported to play an important role in the phenolic composition and sensory characteristics of the wines. After 10 months of bottling, aging odors such as red fruit and caramel showed a decrease in intensity perceived [44].

The color of red wine is one of the main attributes that is perceived by the consumer and may reflect some wine defects [45]. According to the results from Figure 1, the color of wines was not affected by either of the two levels of enrichment. Anthocyanins are the main compounds responsible for the bright red color of wines [43], GPE showed that monomeric anthocyanins were a minor fraction of phenolic compounds. This can explain why enriched wines were similar in color compared with the WC. Copigmentation is the process of color stabilization in wines where anthocyanins and condensed tannins form polymeric pigments. This process can be stimulated by oxygen exposure but can lead to browning reactions and produce protein precipitation that results in a decrease in astringency and bitterness [44,45]. The enrichment of W4 and W8 was performed one week before sensory evaluation. The short period of time could be insufficient to produce stabilization reactions and the modification of color. Brightness was rated in ML intensity for all samples, but different scores were observed (71.1 mm for WC, 50.2 mm for W4, and 55.0 mm for W8), but these differences were not significant (Figure 1). Brightness is defined as the capacity to transmit light into the wine [46]. The quantity of solute dissolved in W4 and W8 can modulate the transmission of light into the wine and explain the difference in scores observed compared with WC.

Bitterness increased in wines with GPE enrichment (Figure 1). This can be explained considering that bitterness is related to monomeric flavan-3-ols on wines, being that epicatechin is more bitter than catechin [47,48]. As observed in the phenolic profile of enriched wines (Table 2), catechin and epicatechin content increased during enrichment, which may explain the observed increase in bitterness. According to Gonzalo-Diago et al. [47], catechin and epicatechin are present in wines in concentrations below the sensory threshold. This may explain why bitterness perceived by judges was not significantly different among wine samples even when the concentration of both monomeric flavan-3-ols increased in W4 and W8. Wine acceptance by consumers generally is assessed when there is a good balance between sourness, bitterness, and other attributes such as astringency. It has been reported that in red wines sugar may reduce bitter taste [49]. In this study, results showed an opposite effect indicating that the slight increase in bitterness and sourness may mask the sweet taste in W4 and W8 samples. Similar results were found previously when an extended maceration time to increase the phenolic content was applied in white wines. This method of enrichment has been shown to have no relationship between the time of maceration with an increase of bitterness in wine but has also shown to decrease sweetness perception. This effect was attributed to the increasing in sourness in the wine [50]. 

Mouthfeel, which represents a key indicator of the sensory characteristics of wine, is defined as a tactile sensation in the oral cavity during consumption. The principal mouthfeel sensations analyzed in wines are astringency, burning, prickling, body, and viscosity [51]. Astringency is defined as a feeling of dryness or roughness produced by tongue friction with the mouth surface because of salivary proteins precipitation [52]. Procyanidins with a high mean degree of polymerization (mDP) and epicatechin-3-gallate have been pointed out as the main ones responsible for astringency perception in red wines [39]. According to the phenolic profile of wine samples (Table 2), enriched red wines presented a higher content of procyanidin B1 and B2, as well as the presence of procyanidins B3, B4, C1, and C2. These changes in phenolic composition and content can explain the differences in astringent perception in wine samples (Figure 1). It has been reported that grape seed content is involved in the astringency of wines due to its higher mDP procyanidins content [39]. The wine body is a complex mouthfeel described in different ways depending on if the consumer is a wine expert or a casual wine consumer. However, it can be defined as the weight of the wine on the palate. The wine body is related to alcohol, sugar, and glycerol content in wines [51]. Results from the sensory analysis (Figure 1) showed that the wine body was rated in L for all samples and no significant differences were observed. This can be explained considering that GPE addition does not alter sugar, glycerol, or alcohol content.

### 3.4. Bioaccessibility of Phenolic Compounds from Enriched Red Wines

To be able to exert their beneficial effect, phenolic compounds must be bioaccessible and bioavailable. In other words, phenolic compounds must be released from the food matrix during the gastrointestinal process in order to be absorbed in the intestine [53]. The effect of the enrichment of red wines on the in vitro bioaccessibility of phenolic compounds was evaluated in three gastrointestinal stages: oral, gastric, and intestinal. The main phenolic compounds released or solubilized in each stage are reported in Table 3. 

The phenolic profile during the digestion in vitro was modified, phenolic compounds were reduced in type and content in the later stages compared to their initial phenolic profile. This change was observed for all samples. Bioaccessibility depends on interactions between phenolic compounds and other types of molecules, such as proteins, carbohydrates, and lipids [54]. Contrary to other food matrices, wine is a liquid matrix where phenolic compounds are dissolved. In this sense, mechanical processes like mastication were not taken into account for this digestive model. Results from Table 3 show that gallic acid content increased during gastric and intestinal stages in the three wine samples. On the other hand, flavan-3-ols decreased in all wine samples, unlike epicatechin that increases only in W8.

The number of phenolic compounds that could be identified in the three stages of the in vitro digestion process decreased in comparison to the number of phenolic compounds in the wine samples (Table 3). A previous study on the bioaccessibility of wine phenolics during in vitro digestion observed that in the oral phase, the phenolic profile of wine was not modified, that is, all the original wine phenolic compounds were detected in the oral stage [55]. The differences can be attributed to incubation times. In the study performed by Sun et al. [55], the oral phase was conducted for 1 min; meanwhile, in the digestive model used in this study, the oral phase was carried out for 5 min. The incubation period has an influence over phenolic compounds since they can interact with salivary proteins such as alpha-amylase and mucin. Phenolic acids can form covalent or non-covalent associations with salivary proteins rich in proline and histidine, depending on their size [56]. Additionally, flavan-3-ols can interact with mucin and proteins rich in proline to generate aggregates and precipitates [57]. The incubation period and interaction with salivary proteins can explain the reduction in the number and content of phenolic compounds in oral and other in vitro digestion stages. Not all of the phenolic compounds content was reduced, and catechin was not modified during the oral phase (Table 3). This effect agrees with those reported by Laurent et al. [58], that reported that catechin and epicatechin presented low affinity with globular proteins such as alpha-amylase. Quercetin-3-glucuronide also presented an increase during the oral phase. This trend has been reported previously for red wines [59]. 

Previous studies on phenolic compound’s bioaccessibility from the wine have shown a general reduction of phenolic compounds during the gastric stage [55,59]. This trend is in accordance with the results observed in the present study. Syringic acid, naringin, catechin, and epicatechin content decreased compared to their initial content in wine (Table 3). This trend was observed for all wine samples. Studies on wines and fruit wines have demonstrated that gastric pH (approximately 1) can influence the stability of phenolic compounds and provoke a loss of them [55,59,60]. 

The instability of phenolic compounds during the gastric stage may explain the loss of phenolic compounds and the increase of phenolic acids such as gallic acid by hydrolysis of octyl-gallate and ellagic acid (Table 3). pH may also influence the content of oligomeric and polymeric procyanidins and promote the formation of monomeric flavan-3-ols such as catechin and epicatechin [61]. Quercetin-3-glucuronide increased its content in all wine samples during the gastric phase. This behavior has been reported previously for other red wines [59]. The increase in quercetin-3-glucuronide during the gastric phase can be attributed to cleavage from proteins or carbohydrates where some flavonols can be found covalently linked in grapes and wines [59,62]. This process also can explain the increase of syringetin-3-glucoside and laricitrin-3-glucoside in wine samples (Table 3). 

The intestinal stage presents the highest enzymatic activity and another drastic pH change due to the sodium carbonate secretion (pH value around 8). Changes in pH value play a major role on phenolic compounds’ stability and reactions that can take place in the intestinal stage [59,63]. Phenolic compounds’ bioaccessibility can be reduced during the intestinal stage due to alkaline conditions [64]. According to the results shown on Table 3, it can be observed that most of the phenolic compounds were not detected in this intestinal stage for the WC sample. A similar trend was observed for the W4 sample. This can be attributed to a change in the pH of the medium from acid to alkaline leading to a change in phenolic compound structure or phenolic degradation. Caffeic acid presented an increase during the intestinal stage, as observed in Table 3. An increase in caffeic acid has been previously reported by Sun et al. [55] and was attributed to chlorogenic acid hydrolysis into caffeic and quinic acid. On the other hand, during copigmentation, anthocyanins can be bonded to phenolic acids such as coumaric and caffeic acid [65]. It is well established that anthocyanins are unstable at pH values higher than 7.5 and can produce the cleavage of phenolic acids such as caffeic acid [66]. 

Other phenolic compounds, such as gallic acid, astilbin, laricitin-3-glucoside, and syringetin-3-glucoside, also presented an increase during the intestinal stage; a similar trend has been observed by other authors [59]. In summary, during their pass through the digestive system, phenolic compounds from wine are exposed to extreme changes in pH or exposed to interact with digestive enzymes and other phenolic compounds, factors that can reduce their bioaccessibility. In contrast some phenolic compounds such as procyanidins, may be liberated from interactions with some wine proteins or produced from the degradation of polymeric forms, hence increasing their bioaccessibility. Therefore, the profile of soluble (bioaccessible) phenolic compounds from wine will change during the gastrointestinal process.

Potential bioavailability of phenolic compounds during intestinal stage from WC, W4, and W8 was determined by diffusion using dialysis membranes, and results can be observed in Table 4.

Only 13 phenolic compounds were detected and quantified after dialysis, fewer compounds than those identified in the three stages of the in vitro digestion process. In WC, only 5 compounds were observed: gallic, protocatechuic, and caffeic acids, naringenin, quercetin-3-glucuronide, and syringetin-3-glucoside. Enriched red wines presented more phenolic compounds, as more compounds were detected in the W8 sample.

Results from the dialysis process (Table 4) showed that few compounds are able to pass into the dialysis bag. A similar trend was previously observed by Lingua et al. [59] where the dialyzable fraction from red wine presented 14 phenolic compounds from 35 compounds detected in gastrointestinal digestion samples. Comparing the samples, W8 presented more phenolic compounds in the dialyzable fraction, mainly hydroxybenzoic acids. This behavior has also been previously observed for red wines and fruit wines [55,59,60]. Using the dialysis method to determine the potentially bioavailability fraction of phenolic compounds from enriched red wine has limitations, mostly because it does not take into account the active transport [67]. Moreover, membrane dialysis is a non-physiological material and molecular interaction that can occur during phenolic absorption is not observable [60]. Nevertheless, the dialysis method offers a good option to screen possible phenolic compounds that can be bioavailable during wine digestion. From this point of view, results showed that W8 increases the quantity of the phenolic compounds that are potentially bioavailable compared to WC. It has been reported previously that the phenolic compounds from wines have low dialyzable percentages around 11 and 37%, and mainly phenolic acids [59,60]. The dialysis method only permits the passing into the dialysis bag of phenolic compounds with low molecular weights, this explains why phenolic acids and flavonoids can be detected in the potentially bioavailable fraction while the presence of more complex phenolic compounds, such as procyanidins, which cannot pass through the dialysis bag, are not detected in the dialyzable fraction. This behavior is consistent with those observed in Table 4. According to results, there are polyphenolic compounds that remain in the non-dialyzable fraction during the intestinal process. These compounds may continue their passage to the colonic portion. In the colonic portion, there are a large variety of microorganisms that can degrade these phenolic compounds and biotransform them into other metabolites that can exhibit a beneficial effect on health [60]. Nevertheless, colonic fermentation was not performed in this study. Further studies must be performed to determine if the non-dialyzable fraction could have an effect during colonic fermentation.

## 4. Conclusions

The present study presents an alternative method to increase the phenolic content of red wines using a phenolic extract from grape pomace, the principal byproduct generated by the winery sector. The phenolic profile of wine was modified by the addition of the GPE, mainly the flavanol and flavan-3-ol fractions. The increment and modification of the phenolic profile led to the modification of two sensory characteristics of wine: astringency and sweetness. No other important characteristics such as color and odor descriptors were modified compared to the control wine. These results may, in a positive way, impact consumer preference for an enriched red wine in contrast to no added wine. Bioaccessibility results showed a decrease in phenolic compounds during the in vitro digestion process. However, enriched red wines presented a higher concentration of phenolic compounds during the intestinal stage, where they would be accessible to be absorbed. Moreover, more individual compounds were detected in the dialyzable fraction of the enriched wine, suggesting that more phenolic compounds can be bioavailable in an enriched wine. On the other hand, results suggest that even when low phenolic compounds are detected in the dialyzable fraction, others can remain non-dialyzable and continue their course to the colonic portion. Further studies should be done to investigate how non-dialyzable phenolic compounds can be modified during colonic fermentation. Still, the bioavailability of enriched red wines in in vivo studies should be investigated. On the other hand, studies on consumer preference for enriched red wines should be considered. 

## Figures and Tables

**Figure 1 foods-12-01194-f001:**
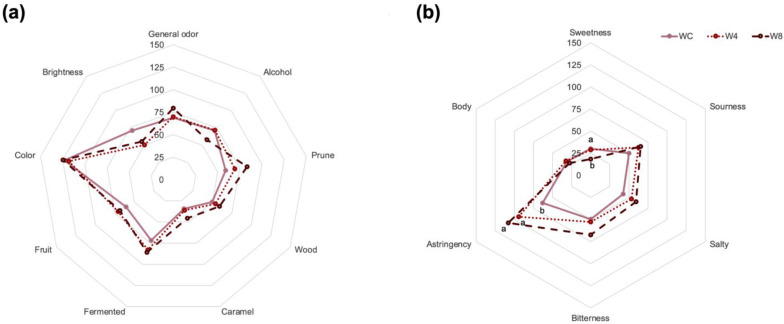
Results from sensory analysis. (**a**) Results from olfactory and sight attributes or descriptors for wine samples; (**b**) Results from taste and mouthfeel attributes from wine samples. Different letters express significant difference (*p* < 0.05).

**Table 1 foods-12-01194-t001:** Phenolic compound characterization of Merlot wine (WC) and Cabernet Sauvignon extract (GPE).

Phenolic Fraction	WC (mg/L)	GPE (mg/g Extract)
Total phenolic content (GAE)	2310.61 ± 25.06	523.75 ± 5.63
Flavonoid content (CE)	547.95 ± 32.04	161.79 ± 10.88
Condensed tannin content (CE)	139.48 ± 9.03	48.08 ± 1.29
Monomeric anthocyanin content (M3E)	23.69 ± 1.84	1.67 ± 0.03

Results express the mean ± SD of three independent experiments. Gallic acid equivalents (GAE). Catechin equivalents (CE). Malvidin-3-glucoside equivalents (M3E).

**Table 2 foods-12-01194-t002:** Phenolic profile of grape pomace extract (GPE) from Cabernet Sauvignon, Merlot wine (WC), and enriched wines at 4.23 (W4) and 7.89 (W8) g GAE/L.

Compound	GPE (mg/g Extract)	WC (mg/L)	W4 (mg/L)	W8 (mg/L)
Hydroxybenzoic acids				
Gallic acid	0.50 ± 0.03	12.05 ± 1.09 ^a^	10.51 ± 0.30 ^a^	4.90 ± 2.81 ^b^
3-methyl-gallic acid	N.D.	<L.O.Q.	<L.O.Q.	<L.O.Q.
4-methyl-gallic acid	<L.O.Q.	N.D.	<L.O.Q.	3.40 ± 0.05
Octyl-gallate	<L.O.Q.	2.06 ± 0.02 ^b^	2.11± 0.04 ^b^	3.90 ± 0.03 ^a^
Pyrogallol	0.34 ± 0.01	N.D.	5.07 ± 0.05 ^a^	5.66 ± 0.37 ^a^
Gentisic acid	N.D.	<L.O.Q.	<L.O.Q.	<L.O.Q.
*p*-hydroxybenzoic acid	0.28 ± 0.01	<L.O.Q.	N.D.	<L.O.Q.
Protocatechuic acid	N.D.	2.58 ± 0.13 ^b^	2.92 ± 0.12 ^b^	6.55 ± 0.40 ^a^
Syringic acid	0.20 ± 0.00	6.20 ± 0.19 ^c^	7.66 ± 0.30 ^b^	16.04 ± 0.10 ^a^
Ellagic acid	0.40 ± 0.00	2.88 ± 0.08 ^c^	3.52 ± 0.07 ^b^	13.56 ± 0.27 ^a^
Vanillic acid	N.D.	<L.O.Q.	<L.O.Q.	<L.O.Q.
Vanillic acid glucoside	<L.O.Q.	N.D.	<L.O.Q.	3.54 ± 0.01
Homovanillic acid	N.D.	2.80 ± 0.17 ^b^	2.54 ± 0.05 ^b^	4.63 ± 0.21 ^a^
Vainilloside	N.D.	N.D.	<L.O.Q.	N.D.
Hydroxycinnamic acids				
Caffeic acid	<L.O.Q.	16.22 ± 0.44 ^b^	16.24 ± 0.43 ^b^	25.27 ± 1.44 ^a^
Caffeoyl-malate	N.D.	<L.O.Q.	N.D.	N.D.
Methyl-ferulate	<L.O.Q.	5.08 ± 0.09 ^b^	4.92 ± 0.10 ^b^	8.94 ± 0.22 ^a^
*m*-coumaric acid	N.D.	<L.O.Q.	<L.O.Q.	<L.O.Q.
*p*-coumaric acid	N.D.	0.59 ± 0.01	<L.O.Q.	<L.O.Q.
Stilbenes				
Resveratrol	N.D.	<L.O.Q.	N.D.	11.32 ± 0.03
Dihydroresveratrol	N.D.	N.D.	N.D.	<L.O.Q.
Resveratrol-glucoside	N.D.	<L.O.Q.	<L.O.Q.	<L.O.Q.
Piceatannol	N.D.	<L.O.Q.	N.D.	<L.O.Q.
Astringin	N.D.	<L.O.Q.	<L.O.Q.	<L.O.Q.
ε-Viniferin	N.D.	N.D.	<L.O.Q.	<L.O.Q.
δ-Viniferin	N.D.	N.D.	N.D.	<L.O.Q.
Resveratrol dimer	N.D.	N.D.	N.D.	<L.O.Q.
Phenylethanoids				
Tyrosol	N.D.	ID	ID	ID
Hydroxytyrosol	N.D.	ID	N.D.	N.D.
Coumarins				
Esculetin	ID	ID	ID	ID
Esculetin glucoside	N.D.	N.D.	ID	N.D.
Flavones				
Luteolin	<L.O.Q.	N.D.	N.D.	<L.O.Q.
Equol	N.D.	N.D.	N.D.	<L.O.Q.
Tetrahydroxyisoflavone	<L.O.Q.	N.D.	N.D.	20.26 ± 0.04
Pentahydroxyisoflavone	<L.O.Q.	N.D.	N.D.	20.39 ± 0.02
Flavanones				
Eriodictyol	0.35 ± 0.00	2.31 ± 0.00 ^b^	2.24 ± 0.05 ^b^	4.51 ± 0.07 ^a^
Eriodictyol-7-glucoside	0.35 ± 0.00	2.45 ± 0.00 ^b^	N.D.	5.01 ± 0.02 ^a^
Homoeriodictyol	N.D.	2.22 ± 0.01	N.D.	N.D.
Hesperetin	N.D.	N.D.	2.21 ± 0.00 ^b^	4.38 ± 0.02 ^a^
Naringenin	1.40 ± 0.10	N.D.	N.D.	48.33 ± 0.86
Naringenin glucoside	N.D.	7.02 ± 0.14 ^b^	N.D.	20.19 ± 1.03 ^a^
Naringin	N.D.	203.88 ± 9.43 ^a^	177.34 ± 5.59 ^b^	219.06 ± 7.55 ^a^
Flavonols				
Astilbin	1.79 ± 0.00	13.26 ± 0.04 ^b^	13.32 ± 0.05 ^b^	25.94 ± 0.11 ^a^
Isorhamnetin	1.83 ± 0.01	<L.O.Q.	11.23 ± 0.01 ^b^	29.96 ± 0.10 ^a^
Isorhamnetin glucoside	1.90 ± 0.01	N.D.	11.84 ± 0.04 ^b^	25.82 ± 0.15 a
Kaempferol	N.D.	N.D.	N.D.	24.07 ± 0.03
Kaempferol-3-glucoside	N.D.	N.D.	10.31 ± 0.05 ^b^	21.42 ± 0.06 ^a^
Kaempferol-7-glucuronide	N.D.	N.D.	N.D.	<L.O.Q.
Laricitrin	N.D.	N.D.	N.D.	23.88 ± 0.08
Laricitrin-3-glucoside	1.87 ± 0.01	11.42 ± 0.02 ^c^	12.12 ± 0.03 ^b^	25.75 ± 0.09 ^a^
Myricetin	<L.O.Q.	N.D.	N.D.	29.06 ± 0.24
Quercetin	1.94 ± 0.03	11.33 ± 0.01 ^b^	11.40 ± 0.01 ^b^	52.89 ± 0.33 ^a^
Quercetin-3-galactoside	2.06 ± 0.02	N.D.	13.20 ± 0.04 ^b^	31.72 ± 0.11 ^a^
Quercetin-3-glucuronide	2.08 ± 0.04	13.35 ± 0.04 ^c^	16.85 ± 0.09 ^b^	40.43 ± 0.33 ^a^
Syringetin	1.78 ± 0.00	<L.O.Q.	<L.O.Q.	24.15 ± 0.05
Syringetin-3-glucoside	2.05 ± 0.03	13.06 ± 0.06 ^c^	14.35 ± 0.08 ^b^	31.85 ± 0.06 ^a^
Taxifolin	1.79 ± 0.00	11.38 ± 0.02 ^c^	11.58 ± 0.06 ^b^	23.86 ± 0.15 ^a^
Flavan-3-ols				
Catechin	1.68 ± 0.13	3.37 ± 0.13 ^c^	9.00 ± 0.17 ^b^	41.50 ± 3.08 ^a^
3-methyl-catechin	N.D.	0.97 ± 0.02 ^b^	N.D.	1.93 ± 0.05 ^a^
Gallocatechin	N.D.	0.70 ± 0.04 ^a^	N.D.	1.16 ± 0.12 ^a^
Epicatechin	0.55 ± 0.04	2.64 ± 0.04 ^c^	7.67 ± 0.11 ^b^	25.11 ± 0.40 ^a^
Epicatechin-3-glucuronide	<L.O.Q.	0.84 ± 0.01 ^b^	0.95 ± 0.05 ^a^	1.82 ± 0.07 ^a^
Epicatechin gallate	<L.O.Q.	N.D.	N.D.	4.93 ± 0.08
Epigallocatechin	N.D.	N.D.	N.D.	<L.O.Q.
Procyanidin B1	0.51 ± 0.10	1.64 ± 0.08 ^b^	2.12 ± 0.16 ^b^	15.49 ± 3.62 ^a^
Procyanidin B2	0.35 ± 0.03	1.81 ± 0.06 ^c^	13.64 ± 0.12 ^b^	45.86 ± 4.82 ^a^
Procyanidin B3	0.98 ± 0.13	N.D.	3.12 ± 0.07 ^a^	6.96 ± 1.69 ^a^
Procyanidin B4	0.17 ± 0.03	N.D.	1.34 ± 0.17 ^a^	3.15 ± 0.79 ^a^
Procyanidin C1	0.44 ± 0.06	N.D.	4.17 ± 0.62 ^b^	19.27 ± 1.51 ^a^
Procyanidin C2	0.40 ± 0.03	N.D.	1.64 ± 0.69 ^b^	5.39 ± 0.62 ^a^
∑ Total phenolic compounds	27.99	354.11	407.13	1033.21

Results express the mean ± SD of three independent experiments. Different letters express significant difference among wine samples at *p* < 0.05. N.D. = not detected compound. I.D. = Identified compound but not quantified. <L.O.Q. = under limit of quantification.

**Table 3 foods-12-01194-t003:** Phenolic compounds detected in each stage of in vitro digestion of WC and enriched wines.

Compound	WC			W4			W8		
	Oral	Gastric	Intestinal	Oral	Gastric	Intestinal	Oral	Gastric	Intestinal
Hydroxybenzoic acids									
Gallic acid	15.7 ± 0.68 ^c^	18.4 ± 0.36 ^b^	35.20 ± 0.62 ^a^	14.19 ± 0.00 ^c^	17.64 ± 2.43 ^b^	35.87 ±0.36 ^a^	13.51 ± 2.97 ^c^	21.20 ± 0.03 ^b^	40.8 ± 0.85 ^a^
Pyrogallol	N.D.	N.D.	N.D.	N.D.	N.D.	N.D.	6.64 ± 0.61 ^c^	14.00 ± 0.39 ^b^	27.07 ± 0.09 ^a^
Protocatechuic acid	4.48 ± 0.10	<L.O.Q.	N.D.	5.14 ± 0.42	<L.O.Q.	<L.O.Q.	6.04 ± 0.04 ^c^	9.14 ± 0.08 ^b^	16.09 ± 0.31 ^a^
Syringic acid	4.79 ± 0.54	N.D.	N.D.	7.85 ± 0.26 ^a^	3.97 ± 1.05 ^b^	N.D.	13.21± 1.34 ^a^	7.59 ± 1.22 ^b^	9.25 ± 0.04 ^b^
Homovanillic acid	4.74 ± 0.04	<L.O.Q.	<L.O.Q.	4.44 ± 0.13	<L.O.Q.	<L.O.Q.	4.59 ± 0.09 ^b^	7.99 ± 0.25 ^a^	<L.O.Q.
Hydroxycinnamic acids									
Caffeic acid	3.62 ± 0.05 ^b^	N.D.	17.68 ± 0.10 ^a^	2.48 ± 0.17 ^b^	N.D.	15.73 ± 0.26 ^a^	5.30 ± 0.13 ^b^	<L.O.Q.	14.65 ± 0.96 ^a^
Flavanones									
Eriodictyol-7-glucoside	5.22 ± 0.01	<L.O.Q.	N.D.	2.27 ± 0.00 ^b^	5.09 ± 0.00 ^a^	N.D.	5.01 ± 0.02 ^c^	5.50 ± 0.02 ^b^	11.12 ± 0.02 ^a^
Naringin	206.01 ± 6.84 ^a^	161.15 ± 1.54 ^b^	163.01 ± 3.69 ^b^	185.33 ± 5.83 ^a^	150.66 ± 5.07 ^b^	156.68 ± 3.22 ^b^	188.13 ± 3.93 ^a^	158.88 ± 2.23 ^b^	154.9 ± 0.77 ^b^
Flavonols									
Astilbin	26.79 ± 0.02 ^a^	N.D.	N.D.	27.67 ± 0.30 ^a^	N.D.	<L.O.Q.	28.07 ± 0.05 ^b^	56.47 ± 0.06 ^a^	N.D.
Isorhamnetin-3-glucoside	N.D.	N.D.	N.D.	26.46 ± 0.03 ^a^	N.D.	N.D.	27.58 ± 0.07 ^b^	56.27 ± 0.06 ^a^	<L.O.Q.
Laricitrin-3-glucoside	N.D.	N.D.	N.D.	26.61 ± 0.02 ^a^	N.D.	N.D.	27.73 ± 0.21 ^c^	56.29 ± 0.02 ^b^	101.0 ± 0.08 ^a^
Quercetin-3-galactoside	N.D.	N.D.	N.D.	27.07 ± 0.02	N.D.	<L.O.Q.	30.19 ± 0.13 ^d^	56.81 ± 0.06 ^b^	101.8 ± 0.03 ^a^
Quercetin-3-glucuronide	26.30 ± 0.02 ^b^	N.D.	102.06 ± 0.15 ^a^	28.78 ± 0.02 ^b^	N.D.	104.52 ± 0.15 ^a^	34.35 ± 0.15 ^c^	56.96 ± 0.04 ^b^	108.7 ±0.18 ^a^
Syringetin-3-glucoside	26.89 ± 0.25 ^c^	56.39 ± 0.03 ^b^	101.51 ± 0.10 ^a^	28.63 ± 0.03 ^b^	<L.O.Q.	102.80 ± 0.15 ^a^	31.43 ± 0.17 ^c^	58.25 ± 0.10 ^b^	104.2 ± 0.17 ^a^
Flavan-3-ols									
Catechin	3.54 ± 0.02	N.D.	N.D.	5.71 ± 0.17 ^b^	N.D.	14.37 ± 0.03 ^a^	13.16 ± 0.27 ^a^	9.59 ± 0.28 ^a^	N.D.
Epicatechin	1.85 ± 0.01	N.D.	<L.O.Q.	3.36 ± 0.17	N.D.	<L.O.Q.	8.97 ± 0.14 ^a^	5.00 ± 0.09 ^c^	6.84 ± 0.30 ^b^
Procyanidin B1	N.D.	N.D.	N.D.	3.23 ± 0.06	N.D.	N.D.	5.42 ± 0.17 ^b^	10.30 ± 0.99 ^a^	N.D.
Procyanidin B2	<L.O.Q.	N.D.	N.D.	10.76 ± 0.26	<L.O.Q.	<L.O.Q.	32.49 ± 0.53 ^a^	16.01 ± 0.83 ^b^	<L.O.Q.
Procyanidin B3	N.D.	N.D.	N.D.	3.60 ± 0.04	N.D.	N.D.	7.10 ± 0.08	N.D.	N.D.
Procyanidin B4	N.D.	N.D.	N.D.	<L.O.Q.	N.D.	N.D.	3.84 ± 0.11 ^a^	<L.O.Q.	N.D.
Procyanidin C1	N.D.	N.D.	N.D.	5.77 ± 0.16 ^a^	N.D.	N.D.	10.24 ± 1.06 ^a^	9.81 ± 0.91 ^a^	<L.O.Q.
Procyanidin C2	N.D.	N.D.	N.D.	5.39 ± 0.62 ^a^	N.D.	N.D.	17.40 ± 3.44 ^a^	8.69 ± 1.77 ^b^	<L.O.Q.

Results express the mean ± SD of three independent experiments. Different letters express significant differences among stages for each wine at *p* < 0.05. Quantified compounds are expressed in mg/L of wine. N.D. = not detected compound. <L.O.Q. = under the limit of quantification.

**Table 4 foods-12-01194-t004:** Potentially bioavailable phenolic compounds.

Compound	WC	W4	W8
Hydroxybenzoic acids			
Gallic acid	3.78 ± 0.09 ^b^	3.68 ± 0.29 ^b^	4.38 ± 0.11 ^a^
3-methyl gallic acid	N.D.	N.D.	<L.O.Q.
Ethyl gallate	N.D.	N.D.	<L.O.Q.
Pyrogallol	N.D.	N.D.	2.91 ± 0.12
p-hydroxybenzoic	N.D.	<L.O.Q.	<L.O.Q.
Protocatechuic acid	<L.O.Q.	<L.O.Q.	1.76 ± 0.00
Syringic acid	N.D.	N.D.	0.79 ± 0.06
Hydroxycinnamic acids			
Caffeic acid	1.78 ± 0.08 ^a^	1.60 ± 0.04 ^b^	1.43 ± 0.07 ^c^
Flavanones			
Naringin	15.80 ± 0.45 ^a^	16.24 ± 0.98 ^a^	14.94 ± 0.64 ^a^
Flavonols			
Quercetin-3-glucuronide	11.24 ± 0.01 ^c^	11.44 ± 0.02 ^b^	11.67 ± 0.02 ^a^
Syringetin 3-glucoside	<L.O.Q.	<L.O.Q.	<L.O.Q.
Flavan-3-ols			
Epicatechin	N.D.	<L.O.Q.	<L.O.Q.

Results express the mean ± SD from three independent experiments. Different letters express significant difference between rows. Quantified compounds are expressed in mg/L of wine. N.D. = not detected compound. <L.O.Q. = under limit of quantification.

## Data Availability

The data are available from the corresponding author.

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
