# Peer review of "Enriched Red Wine: Phenolic Profile, Sensory Evaluation and In Vitro Bioaccessibility of Phenolic Compounds"

_foods, 2023, doi:10.3390/foods12061194_

Round 1

Reviewer 1 Report

The manuscript foods-2197612 deals with the evaluation by a multidisciplinary approach of the effects of wines fortified with grape pomace extracts. The authors measured the sensory characteristics of the fortified wines, their bio-phenols content, and simulated an absorption process in various districts of the digestive apparatus, finally performing an LC/HRMS determination of phenols in the various matrices. I found the work interesting, as it explores the real impact of the food (wine) fortification with polyphenol-rich remnants such as grape pomace o grape seeds. I think the article can be with some minor modifications.

Line 59-60: I find that some more recent and relevant literature could be cited to give a better glimpse into the recent efforts in the field (QingJin et al. Food and Bioproducts Processing 127, 139-151, Brezoiu et al. Food and chemical toxicology, 133, 110787, Di Stefano et al. Sustainability 2022, 14,11).

Line 130 "per liter" of wine not "per litter"

Line  154: even if the calibration levels range has been reported, it seems that LOQ and LOD levels (even if cited in the tables) are not calculated nor reported. These levels should be reported within this section.

Table 2: please report the abbreviation meaning in the table caption.

Line 338: UVC abbreviation (Ultra Violet C probably) has not been reported previously and it should be.

Author Response

Dear Reviewer 1. We wish to thank you for your time to read the manuscript, and your comments to improve the work.

Reviewer 1 (R1): Line 59-60: I find that some more recent and relevant literature could be cited to give a better glimpse into the recent efforts in the field (QingJin et al. Food and Bioproducts Processing 127, 139-151, Brezoiu et al. Food and chemical toxicology, 133, 110787, Di Stefano et al. Sustainability 2022, 14,11).

Authors (A): Thank you for your comment. In order to improve the introduction section, literature suggested was included in the manuscript to offer a better understanding about recent studies focused on grape pomace extraction strategies and characterization.

R1: Line 130 "per liter" of wine not "per litter"

A: Thank you for your correction. Liter was replaced instead of litter and was marked with the change tracker.

R1: Line 154: even if the calibration levels range has been reported, it seems that LOQ and LOD levels (even if cited in the tables) are not calculated nor reported. These levels should be reported within this section.

A: Thank you for your comment. The information about LOQ and LOD determination was included into the manuscript. The values of LOQ and LOD were included as a supplementary material in Table S1.

R1: Table 2: please report the abbreviation meaning in the table caption.

A: Thank you for your comment. The adequate abbreviation was included in the caption of Table 2 and marked with the change tracker.

R1. Line 338: UVC abbreviation (Ultra-Violet C probably) has not been reported previously and it should be.

A: Thank you for your observation. The meaning of UVC (ultraviolet C light) was included in the manuscript and marked with the change tracker.

Reviewer 2 Report

The authors study whether an enrichment of wine in phenolic compounds can improve wine quality or can be an alternative to improve wine health benefits.

*Methods: It is not clear why you use spectrophotometric methods to "characterize" the extract. Mostly, because you are using HPLC afterwards. Spectrophotometric methods are not reliable.

*Results. The pipeline is missing an essential part that is colonic fermentation. It is well known that majority of phenolic compounds are not absorbed in the small intestine but rather are use by gut microbes and then, partially absorbed. So, it is a fundamental bias considering that the whole paper is about polyphenols.

In line 590-593, the authors mention that "only phenolic acids are detected" which is consistent with table 4. Actually is not, quercetin and naringin are not phenolic acids.

Taking this into consideration, other than increasing phenolic compounds in wine, I am not sure about the conclusions. Wine industry, sommeliers, consumers appreciate each wine for their particular composition so I do not think that that is something you can just change, even without considering the economic part. 

Additonally, "increasing benefitial health effects" of an alcoholic drink is essentially wrong.

Author Response

Dear Reviewer 2. Thank you for your time to read the manuscript and all your comments in order to improve the present work.

Reviewer 2 (R2): Methods: It is not clear why you use spectrophotometric methods to "characterize" the extract. Mostly, because you are using HPLC afterwards. Spectrophotometric methods are not reliable.

Authors (A): Thank you for your observation. Spectrophotometric characterization was used as a first approach to wine and grape pomace samples to have a wide understanding about polyphenolic compounds content in both samples. Indeed, spectrophotometric methods are unspecific and for total phenolic compounds determination there are other compounds that can interfere such as amino acids and carbohydrates (Muñoz-Bernal et al., 2017). Still, determination of total phenolic compounds was useful to determine the amount of grape pomace extract to reach the desired enrichment levels. As you clearly comment, HPLC was used after all, this method was performed to determine the specific polyphenolic compounds that were modified.

R2: Results. The pipeline is missing an essential part that is colonic fermentation. It is well known that majority of phenolic compounds are not absorbed in the small intestine but rather are use by gut microbes and then, partially absorbed. So, it is a fundamental bias considering that the whole paper is about polyphenols.

A: Thank you for your comment. The dialysis bag was used to identify the potentially bioavailable polyphenolic compounds during the intestinal stage. As other authors have reported previously (Lingua et al., 2018), small intestinal stage can be coupled with dialysis to observe the dialyzable fraction or the polyphenolic compounds that can be potentially bioavailable during small intestine stage of digestion. We agree that the majority part of polyphenolic compounds can be modified or partially absorbed by microbiota during colonic fermentation and the use of dialysis bag it doesn’t intend to mimic this part. Moreover, we are now working into an in vitro digestion method that includes the colonic fermentation stage to observe the possible phenolic bioavailability during this stage.

R2: In line 590-593, the authors mention that "only phenolic acids are detected" which is consistent with table 4. Actually is not, quercetin and naringin are not phenolic acids.

A: Thank you for your observation. This phrase was modified into the manuscript. As you mentioned, quercetin is a flavonol and, naringin is a flavanone and these compounds were detected also in the possible bioavailable fraction during intestinal stage. The observation was marked with the change tracker.

R2: Taking this into consideration, other than increasing phenolic compounds in wine, I am not sure about the conclusions. Wine industry, sommeliers, consumers appreciate each wine for their particular composition so I do not think that that is something you can just change, even without considering the economic part.

A: Thank you for your comment. Each monovarietal red wine has its specific sensory attributes determined by its phenolic compound content. Each fraction of phenolic compounds affects color, bitter, astringency and other sensory attributes such odor. The authors are aware that when enrichment is performed, we cannot claim that the enriched wine is a monovarietal red wine from Merlot. Even when astringency and sweetness attributes were the only attributes modified from the control wine, the enriched wine is a totally different wine variety. Authors proposed that enriched red wine is more like an ensembled wine between Merlot and Cabernet Sauvignon. We have a work in progress on a sensory consumer test (includes frequently wine drinkers and wine experts) to determine if consumers are willing to drink the enriched red wine or not.

R2: Additionally, "increasing beneficial health effects" of an alcoholic drink is essentially wrong.

A: Thank you for your observation. Alcohol consumption is a matter that have caused strong debate around scientific community. It is well known that alcohol abuse consumption is related to increase the risk of cardiovascular diseases and liver damage (Chiva-blanch & Badimon, 2020; Stránský, 2014). On the other hand, “moderate” consumption of alcohol, one drink per day for women and two drinks per day for men have shown to provide preventive effects against cardiovascular diseases (Chiva-blanch & Badimon, 2020). Among different alcoholic beverages, wine have shown to provide better effects when comparing with spirits or beer (Tverdal et al., 2017). This is attributed mainly to the non-alcoholic fraction (polyphenolic compounds) (Landrault et al., 2003; Shahidi & Ambigaipalan, 2015). This study proposes that a single glass of wine (125 mL) per day, offers 1 g of phenolic compounds (enriched red wine 8 g GAE/L) or 0.5 g of phenolic compounds (enriched red wine at 4 g GAE/L). According to Shahidi & Ambigaipalan, (2015), the estimated world daily ingestion of phenolic compounds is 1 g per day. (Shahidi & Ambigaipalan, 2015). Our proposal is to enrich the wine in order to consume less wine to achieve the daily estimated consumption of phenolic compounds.

References

Chiva-blanch, G., & Badimon, L. (2020). Benefits and risks of moderate alcohol consumption on cardiovascular disease: Current findings and controversies. Nutrients, 12(1). https://doi.org/10.3390/nu12010108

Landrault, N., Poucheret, P., Azay, J., Krosniak, M., Gasc, F., Jenin, C., Cros, G., & Teissedre, P. L. (2003). Effect of a polyphenols-enriched chardonnay white wine in diabetic rats. Journal of Agricultural and Food Chemistry, 51(1), 311–318. https://doi.org/10.1021/jf020219s

Lingua, M. S., Wunderlin, D. A., & Baroni, M. v. (2018). Effect of simulated digestion on the phenolic components of red grapes and their corresponding wines. Journal of Functional Foods, 44(February), 86–94. https://doi.org/10.1016/j.jff.2018.02.034

Muñoz-Bernal, Ó. A., Torres-Aguirre, G. A., Núñez-Gastélum, J. A., de la Rosa, L. A., Rodrigo-Garcia, J., Ayala-Zavala, J. F., & Alvarez-Parrilla, E. (2017). Nuevo acercamiento a la interacción del reactivo de Folin-Ciocalteu con azúcares durante la cuantificación de polifenoles totales. TIP Revista Especializada En Ciencias Químico-Biológicas, 20(2), 28–33.

Shahidi, F., & Ambigaipalan, P. (2015). Phenolics and polyphenolics in foods, beverages and spices: Antioxidant activity and health effects – A review. Journal of Functional Foods, 18, 820–897. https://doi.org/http://dx.doi.org/10.1016/j.jff.2015.06.018

Stránský, M. (2014). Moderate alcohol consumption - Blessing or curse? Kontakt, 16(3), e155–e160. https://doi.org/10.1016/j.kontakt.2014.06.002

Tverdal, A., Magnus, P., Selmer, R., & Thelle, D. (2017). Consumption of alcohol and cardiovascular disease mortality: a 16 year follow-up of 115,592 Norwegian men and women aged 40–44 years. European Journal of Epidemiology, 32(9), 775–783. https://doi.org/10.1007/s10654-017-0313-4

Reviewer 3 Report

Review Report

Manuscript ID: foods-2197612

Title: Enriched red wine: phenolic profile, sensory evaluation and in vitro bioaccessibility of phenolic compounds

This study deals with afterword enrichment of wine (red wine) with phenolic content by addition of grape pomace extract (GPE) into the bottled wine. All experiments are described in detailed and conduct professionally in order to prove or disprove set goes. Merlot red wine was chosen to be enriched with a phenolic extract obtained from Cabernet Sauvignon grape pomace. Two levels of enrichment were assessed, 4 and 8 g/L of total phenolic content (gallic acid equivalents). It was found that the increment of the phenolic profile by afterward modification led to the variation of two sensory characteristics of wine: astringency and sweetness. The bioaccessibility of phenolic compounds from enriched wines was calculated using an in vitro digestive model.  All presented results were discussed at a high scientific level.

My only concern goes in the following direction: what will be consumers’ opinion on the fact that the original wine was modified just before bottling?  

It was a pleasure reading this manuscript; my opinion is that this study has a high potential to be cited. 

Author Response

Dear Reviewer 3. Thank you for your time to read the manuscript. We appreciate all your comments.

Reviewer 3: My only concern goes in the following direction: what will be consumers’ opinion on the fact that the original wine was modified just before bottling? 

Authors: Thank you for your observation. We are aware that the enriched wine cannot be presented as a Merlot wine since their original characteristics from a monovarietal wine were modified. Instead, we though to offer the enriched wine as a new variety of wine or as an ensembled wine of Merlot with Cabernet Sauvignon. We have a work in progress on a sensory consumer test (includes frequently wine drinkers and wine experts) to determine if consumers are willing to drink the enriched red wine or not.

Reviewer 4 Report

The experimental design was reasonable and data were adequate to support the conclusion. Some minor comments were attached below:

Method 2.2: was there any homogenizing processing for the grape pomace samples before enrichment? What's the moisture content? How did the author make sure that the grape pomace sample added to each wine was representative?

Lines 158-165: What are the environmental conditions for enrichment? Please add detailed information about the coffee filters.

Author Response

Dear Reviewer 4. Thank you for your time to read the work and all your comments to improve the manuscript.

Reviewer 4 (R4): Method 2.2: was there any homogenizing processing for the grape pomace samples before enrichment? What's the moisture content? How did the author make sure that the grape pomace sample added to each wine was representative?

Authors (A): Thank you for your comments. The local winery where samples were obtained is small and this allowed us to access to the total production of grape pomace. From the total production, we take 100 kg of grape pomace from different parts of the “cake” to assess a representative sample from the whole production. Once the samples were received in the laboratory facilities, the grape pomace sample was separated into portions to be able to dry them in the oven. The weight of grape pomace was measured every day until constant weight. Data from the drying process is not shown in the manuscript. The total moisture content was 61.51%. After the drying process, the whole grape pomace samples were placed into an industrial mixer. Dried grape pomace was introduced gradually to homogenize the whole sample. Then, grape pomace samples were stored into individual vacuum bags of 1 kg each. These observations were added into the manuscript and marked with the change tracker.

R4: Lines 158-165: What are the environmental conditions for enrichment? Please add detailed information about the coffee filters.

A: Thank you for your comment. The environmental conditions of wine enrichment were added into the manuscript. In brief, enrichment was performed under laboratory conditions at 25°C and light absence. Use of nitrile gloves and face mask was mandatory for the wine enrichment. Information about the coffee filters was added into the manuscript. This observation were marked with the change tracker.

Round 2

Reviewer 2 Report

The authors have commented on my concenrs, however, they are hardly solvable unless they at least include a fermentation step. 

Additionally, whether moderate consumption of wine may have benefits, does not justify encouraging its consumption. Basically because you can get all those benefits and more with other foods that do not include alcohol in their composition. 

Author Response

Dear Reviewer, thank you for your comments. We appreciate your time to read the manuscript to improve it.

Reviewer 2: The authors have commented on my concerns; however, they are hardly solvable unless they at least include a fermentation step.

Authors: Thank you for your comment. We have discussed your concern about including the colonic fermentation step in the in vitro digestive model. As you rightly commented before, in this step, most of the phenolic compounds undergo modifications or degradation by microbiota and can be more readily for absorption. Nevertheless, we were more interested on observe the effect of the upper section of the digestive model over phenolic compounds from wine than colonic fermentation. Since previous study on the effect of simulated digestion on phenolic compounds from wine (Lingua et al., 2018), reported a decrease of phenolic content during mouth, stomach, and upper intestine digestion. We want to determine if enrichment could attenuate this decrease during these steps of the upper digestion system. Finally, we want to make clear that in this manuscript, the idea of phenolic compounds can pass through the colonic part (non-dialyzable fraction) is just a hypothesis that we can´t support with the data obtained. This was stated in the manuscript (lines 722-730). Moreover, we have analyzed the idea of evaluating the effect of mouth, gastric, small intestine, and colonic fermentation over phenolic compounds from enriched red wine. At the present time, we are developing a clinical trial to determine the bioavailable fraction of phenolic compounds from enriched red wine.

Reviewer 2: Additionally, whether moderate consumption of wine may have benefits, does not justify encouraging its consumption. Basically, because you can get all those benefits and more with other foods that do not include alcohol in their composition.

Authors: Thank you for your comment. Authors agree with you. We also discourage the binge consumption of red wine or any other alcoholic beverage. Instead, it has been established that moderate consumption is about 1 glass of wine (125 mL) per day. We are also aware that there are other foods without alcohol that can provide phenolic compounds, for example, grape juice. Nevertheless, even when the content of phenolic compounds can be similar on those found in wine, is a highly sugary drink. On the other hand, complex food matrix like grapes can interfere with the bioavailable fraction. There is diverse literature that has reviewed the different food matrix factors that can affect phenolic bioavailability (Bohn, 2014; Bohn et al., 2015; Ma & Chen, 2020). Finally, as mentioned before, we discourage the abuse of alcohol consumption. Instead, we propose moderate wine consumption to help reach the daily recommended ingestion of phenolic compounds.

References

Bohn, T. (2014). Dietary factors affecting polyphenol bioavailability. Nutrition Reviews, 72(7), 429–452. https://doi.org/10.1111/nure.12114

Bohn, T., Mcdougall, G. J., Alegría, A., Alminger, M., Arrigoni, E., Aura, A. M., Brito, C., Cilla, A., El, S. N., Karakaya, S., Martínez-Cuesta, M. C., & Santos, C. N. (2015). Mind the gap-deficits in our knowledge of aspects impacting the bioavailability of phytochemicals and their metabolites-a position paper focusing on carotenoids and polyphenols. Molecular Nutrition and Food Research, 59(7), 1307–1323. https://doi.org/10.1002/mnfr.201400745

Lingua, M. S., Wunderlin, D. A., & Baroni, M. v. (2018). Effect of simulated digestion on the phenolic components of red grapes and their corresponding wines. Journal of Functional Foods, 44(February), 86–94. https://doi.org/10.1016/j.jff.2018.02.034

Ma, G., & Chen, Y. (2020). Polyphenol supplementation benefits human health via gut microbiota: A systematic review via meta-analysis. Journal of Functional Foods, 66(January). https://doi.org/10.1016/j.jff.2020.103829